# POST-HOC CONCEPT BOTTLENECK MODELS

**Mert Yuksekgonul, Maggie Wang, James Zou**
Stanford University
{merty,maggiewang,jamesz}@stanford.edu

## ABSTRACT

Concept Bottleneck Models (CBMs) map the inputs onto a set of interpretable concepts ("the bottleneck") and use the concepts to make predictions. A concept bottleneck enhances interpretability since it can be investigated to understand what concepts the model "sees" in an input and which of these concepts are deemed important. However, CBMs are restrictive in practice as they require dense concept annotations in the training data to learn the bottleneck. Moreover, CBMs often do not match the accuracy of an unrestricted neural network, reducing the incentive to deploy them in practice. In this work, we address these limitations of CBMs by introducing Post-hoc Concept Bottleneck models (PCBMs). We show that we can turn any neural network into a PCBM without sacrificing model performance while still retaining the interpretability benefits. When concept annotations are not available on the training data, we show that PCBM can transfer concepts from other datasets or from natural language descriptions of concepts via multimodal models. A key benefit of PCBM is that it enables users to quickly debug and update the model to reduce spurious correlations and improve generalization to new distributions. PCBM allows for global model edits, which can be more efficient than previous works on local interventions that fix a specific prediction. Through a model-editing user study, we show that editing PCBMs via concept-level feedback can provide significant performance gains without using data from the target domain or model retraining. The code for our paper can be found in https://github.com/mertyg/post-hoc-cbm.

## 1 INTRODUCTION

There is growing interest in developing deep learning models that are interpretable and yet still flexible. One approach is concept analysis (Kim et al., 2018), where the goal is to understand if and how high-level human-understandable features are "engineered" and used by neural networks. For instance, we may like to probe a skin lesion classifier to understand if the *Irregular Streaks* concept is encoded in the embedding space of the classifier and used later to make the prediction.

Our work builds on the earlier idea of concept bottlenecks, specifically Concept Bottleneck Models (CBMs) (Koh et al., 2020). Concept bottlenecks are inspired by the idea that we can solve the task of interest by applying a function to an underlying set of human-interpretable concepts. For instance, when trying to classify whether a skin tumor is malignant, dermatologists look for different visual patterns, e.g. existence of *Blue-Whitish Veils* can be a useful indicator of melanoma (Menzies et al., 1996; Lucieri et al., 2020). CBMs train an entire model in an end-to-end fashion by first predicting concepts (e.g. the presence of *Blue-Whitish Veils*), then using these concepts to predict the label.

By constraining the model to only rely on a set of concepts and an interpretable predictor, we can: **explain** what information the model is using when classifying an input by looking at the weights/rules in the interpretable predictor and **understand** when the model made a particular mistake due to incorrect concept predictions.

While CBMs provide several of the benefits mentioned above, they have several key limitations:

1. **Data:** CBMs require access to concept labels during model training, i.e. training data should be annotated with which concepts are present. Even though there are a number of densely annotated datasets such as CUB (Wah et al., 2011), this is particularly restrictive for real-world use cases, where training datasets rarely have concept annotations.

2. **Performance:** CBMs often do not match the accuracy of an unrestricted model, potentially reducing the incentive to use them in practice. When the concepts are not enough to solve the desired task, it is not clear how to improve the CBM and match the original model performance, while retaining the interpretability benefits.

3. **Model editing**: Koh et al. (2020) discuss intervening on the model to fix the prediction for a single input, yet it is not shown how to holistically edit and improve the model itself. Intervening only changes the model behavior for a single sample, but global editing changes the model behavior completely. When the model picks up an unintended cue, or learns spurious associations, using the latter approach and editing the concept bottleneck can improve the model performance more generally than an intervention tailored toward one specific input. Prior work on CBMs does not discuss how to globally edit a model's behavior. Ideally, we would like to edit models with the help of human input in order to lower computational costs and remove assumptions about data access.

**Our contributions.** In this work, we propose the **Post-hoc Concept Bottleneck Model (PCBM)** to address these important challenges. PCBMs can convert any pre-trained model into a concept bottleneck model in a data-efficient manner, and enhance the model with the desired interpretability benefits. When the training data does not have concept annotations, which is often the case, PCBM can flexibly leverage concepts annotated in other datasets and natural language descriptions of concepts. When applicable, PCBMs can remove the laborious concept annotation process by leveraging multimodal models to obtain concept representations; this results in richer and more expressive bottlenecks using natural language descriptions of a concept, making PCBMs more accessible in various settings. Furthermore, when the available concepts are not sufficiently rich, we introduce a residual modeling step to the PCBM to recover the original blackbox model's performance. In experiments across several tasks, we show that PCBMs can be used with comparable performance compared to black-box models. While prior work (Koh et al., 2020) demonstrated the possibility of performing local model interventions to change individual predictions, here we propose interventions for changing global model behavior. Through user studies, we show that PCBM enables efficient **global model edits without retraining or access to data from the target domain** and that users can improve PCBM performance by using concept-level feedback to drive editing decisions.

## 2    POST-HOC CONCEPT BOTTLENECK MODELS

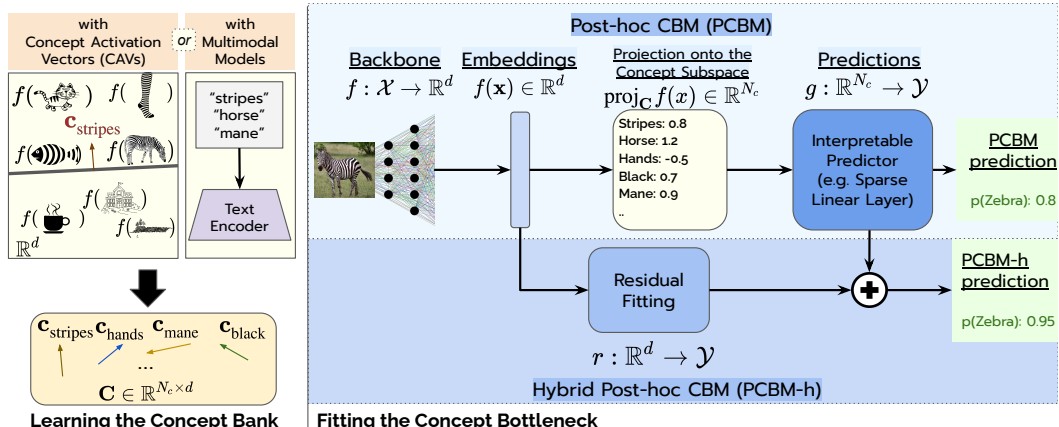

Figure 1: **Post-hoc Concept Bottleneck Models.** First, we learn the vectors in our concept bank. With the CAV approach, for each concept, e.g. stripes, we train a linear SVM to distinguish the embeddings of examples that contain the concept and use the vector normal to the boundary (CAV). When annotations are hard to obtain, we can leverage multimodal models and use the text encoder to map each concept to a vector. Next, we project the embeddings produced by the backbone onto the concept subspace defined by the set of vectors. We then train an interpretable predictor to classify the examples from their projections. When the concept library is incomplete, we can construct a PCBM-h by sequentially introducing a residual predictor that maps the embeddings to the target space.

There are two main steps to building a PCBM. We let $f : \mathcal{X} \to \mathbb{R}^d$ be any pretrained backbone model, where $d$ is the size of the corresponding embedding space and $\mathcal{X}$ is the input space. For instance, $f$ can be the image encoder of CLIP (Radford et al., 2021) or the model layers up to the penultimate layer of a ResNet (He et al., 2016). An overview of PCBMs can be found in 1, and we describe the steps in detail below.

**Learning the Concept Subspace ($C$):** To learn concept representations, we make use of CAVs (Concept Activation Vectors) (Kim et al., 2018). In particular, we first define a concept library $I = \{i_1, i_2, ..., i_{N_c}\}$, where $N_c$ denotes the number of concepts. The concepts in the library can be selected by a domain expert or learned automatically from the data (Ghorbani et al., 2019; Yeh et al., 2020). For each concept $i$, we collect embeddings for the positive examples, denoted by the set $P_i = \{f(x_{p_1}), ..., f(x_{p_{N_p}})\}$, that exhibit the concept, and negative examples $N_i = \{f(x_{n_1}), ..., f(x_{p_{N_n}})\}$ that do not contain the concept. In practice, densely annotated datasets (Caesar et al., 2018; Fong & Vedaldi, 2018) are used to collect points that are positive/negative for the concept. Importantly, unlike CBMs, these samples can be different from the data used to train the backbone model. Following (Kim et al., 2018), we train a linear SVM using $P_i$ and $N_i$ to learn the corresponding CAV, that is, the vector normal to the linear classification boundary. We denote the CAV for concept $i$ as $c_i$. Let $C \in \mathbb{R}^{N_c \times d}$ denote the matrix of concept vectors, where each row $c_i$ represents the corresponding concept $i$. Given an input, we project the embedding of the input onto the subspace spanned by concept vectors (the concept subspace). Particularly, we let $f_C(x) = \text{proj}_C f(x) \in \mathbb{R}^{N_c}$, where the ith entry is $f_C^{(i)}(x) = \frac{\langle f(x), c_i \rangle}{\|c_i\|_2^2} \in \mathbb{R}$. It is important to observe that we do not need to annotate the training data with concepts. Namely, the dataset used to learn concepts can be different from the original training data. In several of our experiments, we use held-out datasets to learn the concept subspace.

**Leveraging multimodal models to learn concepts:** Annotating images with concepts is a laborious process. In practice, this can be an important roadblock before using CBMs. To address this, here we show that we can also leverage natural language concept descriptions and multimodal models to implement concept bottlenecks. This approach alleviates the need of collecting labeled data to construct the concept subspace. Multimodal models such as CLIP (Radford et al., 2021) have a text encoder $f_{\text{text}}$ along with an image encoder, which maps a description to the shared embedding space. We leverage the text encoder to augment the process of learning concept vectors. For instance, we can have $c_{\text{stripes}}^{\text{text}} = f_{\text{text}}(\text{"stripes"})$ where the concept vector for stripes is obtained by mapping the prompt using the text encoder. For each concept description, we can collect the text embeddings and construct our multimodal concept bank $C^{\text{text}}$ as our subspace.

For a given classification task, we can use ConceptNet (Speer et al., 2017) to obtain concepts that are relevant to these classes. ConceptNet is an open knowledge graph, where we can find concepts that have particular relations to a query concept. For instance, we can find relations of the form "A Cat *has* {whiskers, four legs, sharp claws, ..}". Similarly, we can find "parts" of a given class (e.g. "bumper", "roof" for "truck"), or the superclass of a given class (e.g. "animal", "canine" for "dog"). We restrict ourselves to five sets of relations for each class: the *hasA*, *isA*, *partOf*, *HasProperty*, *MadeOf* relations in ConceptNet. We collect all concepts that have these relations with classes in each classification task to build the concept subspace.

**Learning the Interpretable Predictor**: Next, we define an interpretable predictor that maps the concept subspace to the model prediction. Concretely, let $g : \mathbb{R}^{N_c} \to \mathcal{Y}$ be an interpretable predictor, such as a sparse linear model or a decision tree, where $\mathcal{Y} = \{1, 2, ..., K\}$ denotes the label space. An interpretable predictor is desirable because it provides insight into which concepts the model is relying on when making a decision. If a domain expert observes a counter-intuitive phenomenon in the predictor, they can edit the predictor to improve the model. To learn the PCBM, we solve the following problem:

$$\min_{g} \mathbb{E}_{(x,y) \sim \mathcal{D}}[\mathcal{L}(g(f_C(x)), y)] + \frac{\lambda}{N_c K}\Omega(g) \tag{1}$$

where $f_C = \text{proj}_C f(x)$ is the projection onto the concept subspace, $\mathcal{L}(\hat{y}, y)$ is a loss function such as cross-entropy loss, $\Omega(g)$ is a complexity measure to regularize the model, and $\lambda$ is the regularization strength. Note that the concept subspace is fixed during PCBM training. In this work, we use sparse linear models to learn the interpretable predictor, where $g(f_C(x)) = w^T f_C(x) + b$. We apply the softmax function to $g$ if the problem is a classification problem. Similarly, we define

$\Omega(g) = \alpha||\boldsymbol{w}||_1 + (1-\alpha)||\boldsymbol{w}||_2^2$ to be the elastic-net penalty parameterized by $\alpha$, then normalize by the number of classes and concepts.

The expressiveness of the concept subspace is crucial to PCBM performance. However, even when we have a rich concept subspace, concepts may not be enough to solve a task of interest. When the performance of the PCBM does not match the performance of the original model, users have less incentive to use interpretable models like concept bottlenecks. Next, we aim to address this limitation.

**Recovering the original model performance with residual modeling:** What happens when the concept bank is not sufficiently expressive, and PCBM performs worse than the original model? For instance, there may be skin lesion descriptors that are not available in the concept library. Ideally, we would like to preserve the original model's accuracy while retaining the interpretability benefits. Drawing inspiration from the semiparametric model literature on fitting residuals (Härdle et al., 2004), we introduce **Hybrid Post-hoc CBMs (PCBM-h)**. After fixing the concept bottleneck and the interpretable predictor, we re-introduce the embeddings to 'fit the residuals'. In particular, we solve the following:

$$\min_r \mathbb{E}_{(x,y)\sim\mathcal{D}}[\mathcal{L}(g(f_{\boldsymbol{C}}(x)) + r(f(x)), y)] \tag{2}$$

where $r : \mathbb{R}^d \to \mathcal{Y}$ is the residual predictor. Note that in Equation 2, while training the residual predictor, the trained concept bottleneck (the concept subspace ($f_C$) and the interpretable predictor($g$)) is kept fixed, i.e. fitting PCBM-hs is a sequential procedure. We hypothesize that the residual predictor will compensate for what is missing from the concept bank and recover the original model's accuracy. We implement the residual predictor as a linear model, i.e. $r(f(x)) = w_r^T f(x) + b_r$. Given a trained PCBM-h, if we would like to observe model performance in the absence of the residual predictor, we can simply drop $r$ (in other words, concept contributions do not depend on the residual step).

## 3 EXPERIMENTS

We evaluate the PCBM and PCBM-h in challenging image classification and medical settings, demonstrating several use cases for PCBMs. We further address practical concerns and show that PCBMs can be used without a loss in the original model performance. We used the following datasets to systematically evaluate the PCBM and PCBM-h:

**CIFAR10, CIFAR100 (Krizhevsky et al., 2009)** We use CLIP-ResNet50 (Radford et al., 2021) as the backbone model. For the concept bottleneck, we use 170 concepts introduced in (Abid et al., 2022) which are extracted from the BRODEN visual concepts dataset (Fong & Vedaldi, 2018). These include objects (e.g. *dog*), settings, (e.g. *snow*) textures (e.g. *stripes*), and image qualities (e.g. *blurriness*). The full list of concepts can be found in (Abid et al., 2022).

**COCO-Stuff (Caesar et al., 2018)** is a dataset derived from MS-COCO (Lin et al., 2014), consisting of scenes with various object annotations. Previous work (Singh et al., 2020) has shown that there are severe co-occurrence biases in this dataset. For instance, images from the *wine glass* often also have a *dining table* in the image. Singh et al. (2020) identifies these 20 classes with heavy co-occurrence biases and we train PCBMs to recognize these 20 objects, where we minimize the binary cross-entropy loss individually for each class. In this scenario, we again use CLIP-ResNet50 (Radford et al., 2021) as the backbone and BRODEN visual concepts as the set of concepts.

**CUB (Wah et al., 2011)** In the 200-way bird identification task, we use a ResNet18 (He et al., 2016) trained on the CUB dataset[1]. We use the same training/validation splits and 112 concepts as in (Koh et al., 2020). These concepts include *wing shape*, *back pattern*, and *eye color*.

**HAM10000 (Tschandl et al., 2018)** is a dataset of dermoscopic images, which contain skin lesions from a representative set of diagnostic categories. The task is detecting whether a skin lesion is benign or malignant. We use the Inception (Szegedy et al., 2015) model trained on this dataset, which is available from (Daneshjou et al., 2021). Following the setting in (Lucieri et al., 2020), we collect concepts from the Derm7pt (Kawahara et al., 2018) dataset. The 8 concepts obtained from this dataset include *Blue Whitish Veil*, *Pigmented Networks*, *Regression Structures*, which are reportedly associated with the malignancy of a lesion.

---

[1]The CUB pretrained model is obtained from `https://github.com/osmr/imgclsmob`

**SIIM-ISIC (Rotemberg et al., 2021)** To test a real-world transfer learning use case, we evaluate the model trained on HAM10000 on a subset of the SIIM-ISIC Melanoma Classification dataset. We use the same concepts described in the HAM10000 dataset.

We use linear probing on top of the backbones for CIFAR, COCO, and ISIC to obtain the baseline model performance. For CUB and HAM10000, we use the out-of-the-box models trained on the respective datasets without any additional training. We emphasize that in all of our experiments except CUB, we learn concepts using a held-out dataset. Namely, while the original CBMs need to have concept annotations for training images, we remove this limitation by post-hoc learning of concepts with any dataset. To learn the concept subspace, we use 50 pairs of images for each concept to train a Linear SVM. We refer the reader to the Appendix for further details on datasets and training.

**PCBMs achieve comparable performance to the original model:** In Table 1, we report results over these five datasets. PCBM achieves comparable performance to the original model in all datasets except CIFAR100, and PCBM-h matches the performance in all scenarios. Strikingly, PCBMs match the performance of the original model in HAM10000 and ISIC, *with as few as 8 human-interpretable concepts*. In CIFAR100, we hypothesize that the concept bank available is not sufficient to classify finer-grained classes, and hence there is a performance gap between the PCBM and the original model. When the concept bank is not sufficient to solve the task, PCBM-h can be introduced to recover the original model performance while retaining the benefits of PCBM (see next sections). **Comparing to CBM:** A comparison to CBM was not possible in most datasets, as CBMs cannot be trained due to the lack of dense annotations. To compare the data efficiency of both methods, we report a comparison of CBM to PCBM when trained on CUB in Appendix C, finding that CBM can only reach a similar performance to PCBM with *20x* more annotations. **Analyzing the residual predictor:** To understand whether PCBM-h overrides PCBM predictions, in Appendix B, we look at the consistency between PCBM and PCBM-h predictions. We show that the residual component in PCBM-h intervenes only when the prediction is wrong, and fixes mistakes. When PCBM is confident, PCBM-h does not modify the prediction or significantly increase the confidence. In general, the residual component may dominate when the bottleneck is insufficient and future work can aim to explicitly limit the residual component, e.g. PIE (Wang et al., 2021) regularizer.

Table 1: **PCBMs achieve comparable performance to the original model**. We report performance over different scenarios for the original model and PCBMs with concept datasets. In CIFAR100, PCBM performs poorly since the concept bank is not expressive enough to solve a finer-grained task; however, PCBM-h recovers the original model's accuracy. Strikingly, PCBMs match the performance of the original model in HAM10000 and ISIC, *with as few as 8 human-interpretable concepts*. Original CBMs cannot be trained on CIFAR/HAM10000/ISIC/COCO-Stuff, as they do not have concept labels in the training dataset. The mean and standard errors are reported over 10 random seeds. We report AUROC for HAM10000 and ISIC, mAP for COCO-Stuff, and accuracy for CIFAR and CUB.

|  | CIFAR10 | CIFAR100 | COCO-Stuff | CUB | HAM10000 | ISIC |
|---|---|---|---|---|---|---|
| Original Model | 0.888 | 0.701 | 0.770 | 0.612 | 0.963 | 0.821 |
| PCBM | $0.777 \pm 0.003$ | $0.520 \pm 0.005$ | $0.741 \pm 0.002$ | $0.588 \pm 0.008$ | $0.947 \pm 0.001$ | $0.736 \pm 0.012$ |
| PCBM-h | $0.871 \pm 0.001$ | $0.680 \pm 0.001$ | $0.768 \pm 0.01$ | $0.610 \pm 0.010$ | $0.962 \pm 0.002$ | $0.801 \pm 0.056$ |

**PCBM using CLIP concepts:** Here, we show that when labeled examples to learn concepts are not available, we can use multimodal representations such as CLIP to generate concepts without the laborious annotation cost. Using ConceptNet, we obtain concept descriptions for CIFAR10, CIFAR100, and COCO-Stuff tasks (206, 527, and 822 concepts, respectively), using the relations described in Section 2. The concept subspace is obtained by using text embeddings for the concept descriptions generated by the ResNet50 variant of CLIP.

Table 2: **Concept Bottlenecks with CLIP concepts**. When a concept bank is not available or is insufficient, we can use natural language descriptions of concepts with CLIP to implement CBMs.

|  | CIFAR10 | CIFAR100 | COCO-Stuff |
|---|---|---|---|
| Original Model (CLIP) | 0.888 | 0.701 | 0.770 |
| PCBM & labeled concepts | $0.777 \pm 0.003$ | $0.520 \pm 0.005$ | $0.741 \pm 0.002$ |
| PCBM & CLIP concepts | $0.833 \pm 0.003$ | $0.600 \pm 0.003$ | $0.755 \pm 0.001$ |
| PCBM-h & CLIP concepts | $0.874 \pm 0.001$ | $0.691 \pm 0.006$ | $0.769 \pm 0.001$ |

In Table 2, we give an overview of the results. We observe that with a more expressive multimodal

bottleneck, we can almost recover the original model accuracy in CIFAR10, and we reduce the performance gap in CIFAR100. CLIP contains extensive knowledge of natural images, and thus captures natural image concepts (e.g. *red*, *thorns*). However, CLIP is less reliable at representing more granular or domain-specific concepts (e.g. *blue whitish veils*) and hence this methodology is less applicable to more domain-specific applications. While there is a recent trend in training multimodal models for biomedical applications (Zhang et al., 2020; EleutherAI), these models are trained with smaller datasets and are less competitive. We hope that as these models improve, PCBMs will make it easier to deploy concept bottlenecks in more specialized domains.

**Explanations in PCBMs:** In Figure 2, we provide sample concept weights in the corresponding PCBMs. For instance, in HAM10000, PCBMs use *Blue Whitish Veils*, *Atypical Pigment Networks*, and *Irregular Streaks* to identify malignant lesions, whereas *Typical Pigment Networks*, *Regular Streaks*, and *Regular Dots and Globules* are used to identify benign lesions. These associations are consistent with medical knowledge (Menzies et al., 1996; Lucieri et al., 2020). We observe similar cases in CIFAR-10 and CIFAR-100, where the class Rose is associated with the concepts *Flower*, *Thorns* and *Red*. In Appendix Section F, we further analyze the global explanations of COCO-Stuff and show that PCBM reveals co-occurrence biases in the training data. Note that the concept weights for PCBM-h are exactly the same as for PCBM since the residual model is fit after the concept weights are fixed. However, given the addition of the residual component, the net effect of concepts may be different. We also note that the concept weights discussed may not reflect the causal effect. Estimating the causal effect of a concept requires careful treatment (Goyal et al., 2019), e.g. accounting for the interactions between concepts, and we leave this to future work.

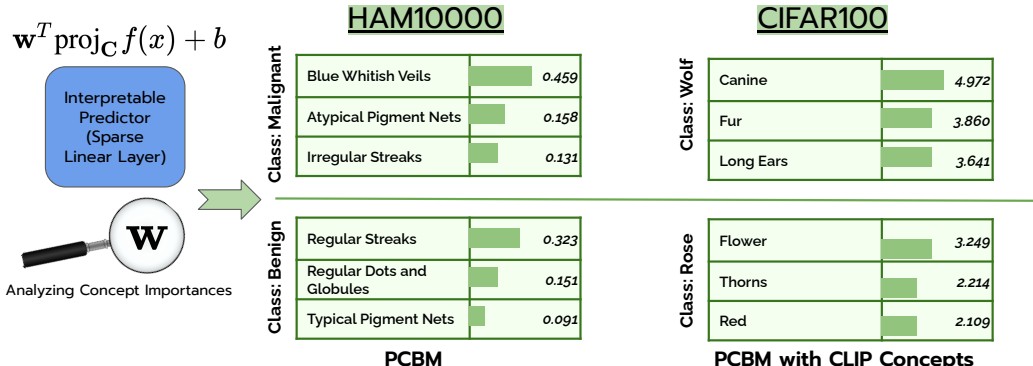

Figure 2: **Explaining Post-hoc CBMs.** We report the top 3 largest weights in the linear layer for the shown classes. For instance, *Blue Whitish Veils*, *Atypical Pigment Networks*, and *Irregular Streaks* have large weights for classifying whether a skin lesion is malignant. These are consistent with dermatologists' findings (Menzies et al., 1996).

## 4 MODEL EDITING WITH POST-HOC CONCEPT BOTTLENECKS

When we observe a counter-intuitive phenomenon or a spurious correlation in the concept bottleneck, can we make the model perform better by simple edit operations? In this section, we show that PCBMs come with the benefit of easy and global concept-level model feedback. Koh et al. (2020) demonstrate the ability of CBMs to incorporate local interventions. Namely, if a user identifies a misprediction in the concept bottleneck, they can intervene on the concept prediction, and change the prediction for the particular instance, we call this a local intervention or edit. Here, we are interested in editing models to perform 'global' edits. Namely, with simple concept-level feedback, we would like to edit a PCBM and change the entire model behavior. For instance, in our experiments, we investigate the use case of adapting a PCBM to a new distribution. Unlike most existing model editing approaches, *we do not need any data from either the training or target domains to perform the model edit*, which can be a significant advantage in practice when data is inaccessible. Given a trained PCBM, we edit our concept bottleneck by manipulating the concept weights. For simplicity, in the experiments below we only focus on positive weights in the linear model.

## 4.1 Controlled experiment: editing PCBM when the spurious concept is known

For our editing experiments, we use the Metashift (Liang & Zou, 2022) dataset to simulate distribution shifts. We use 10 different scenarios where there is a distribution shift for a particular class between the training and test datasets. For instance, during training, we only use table images that also contain a dog in the image and test the model with table images where there is not a dog in the image. We denote the training domain as *table(dog)*. We give more details and the results of all domains in the Appendix. We use an ImageNet pretrained ResNet50 variant as a backbone and the visual concept bank described in the Experiments section. Given a PCBM, we evaluate three **editing strategies**:

1. **Prune**: We set the weight of a particular concept on a particular class prediction to 0, i.e. for a concept indexed by $i$, we let $\tilde{w}_i = 0$.
2. **Prune+Normalize**: After applying pruning, we rescale the concept weights. Let $P$ denote the indices of positive weights that are pruned, $\tilde{P}$ denote the indices of weights that remain, and $\boldsymbol{w}_P, \boldsymbol{w}_{\tilde{P}}$ be corresponding weight vectors for the particular class. We rescale each weight to match the original norm of the vector by letting $\forall i \in \tilde{P}, \ \tilde{w}_i = w_i(1 + \frac{||\boldsymbol{w}_P||_1}{||\boldsymbol{w}_{\tilde{P}}||_1})$, leading to $||\tilde{\boldsymbol{w}}||_1 = ||\boldsymbol{w}||_1$. The normalization step alleviates the imbalance between class weights upon pruning concepts with large weights for a particular class.
3. **Fine-tune (Oracle)**: We compare our simple editing strategies to fine-tuning on the test domain, which can be considered an oracle. Particularly, we fine-tune the PCBM using samples from the test domain and then test the fine-tuned model with a set of held-out samples.

In the context of Metashift experiments, we simply edit the concept spuriously correlated with a particular class. Concretely, for the domain *table(dog)*, we prune the weight of the *dog* concept for the class *table*. In Table 3, we report the result of our editing experiments over 10 scenarios. We observe that for PCBMs, the Prune+Normalize strategy can recover almost half of the accuracy gains achieved by fine-tuning on the original test domain. Further, we observe that PCBM-h obtains lower accuracy gains with model editing. In PCBM, we can remove the concept from the model, but in PCBM-h, there may still be leftover information about the spurious concept in the residual part of the model. This is in line with our expectations, i.e. PCBM-h is a 'less' interpretable but more powerful variant of PCBM. While it still offers partial editing improvements that would not be possible with the vanilla model, it does not bring as much improvement as the PCBM.

Table 3: **Model edits with Post-Hoc CBMs.** We report results over 10 distribution shift experiments generated using Metashift. Accuracy and edit gains are averaged over 10 scenarios and are reported as *mean ± standard error*. We observe that very simple editing strategies in the concept subspace provide almost $50\%$ of the gains made by fine-tuning on the test domain.

|  | Unedited | Prune | Prune + Normalize | Fine-tune (Oracle) |
|---|---|---|---|---|
| PCBM Accuracy | $0.656 \pm 0.025$ | $0.686 \pm 0.026$ | $0.750 \pm 0.019$ | $0.859 \pm 0.028$ |
| PCBM Edit Gain | — | 0.029 | 0.093 | 0.202 |
| PCBM-h Accuracy | $0.657 \pm 0.039$ | $0.672 \pm 0.033$ | $0.713 \pm 0.027$ | $0.861 \pm 0.032$ |
| PCBM-h Edit Gain | — | 0.017 | 0.058 | 0.190 |

Even though our edit strategy is extremely simple, we can recover almost half of the gains made by fine-tuning the model. It is particularly easy to use since it can be applied without fine-tuning or using any knowledge or data from the target domain. However, this methodology requires knowledge of the spurious concept in the training domain. This may not be realistic in practice, and in the next section, we turn to human users for guidance.

## 4.2 User study: editing PCBM with human guidance

One of the advantages of pruning is that it is naturally amenable to bringing humans into the loop. Rather than attempting to automate the edits, we can rely on a human to pick concepts that make logical sense to prune. We show that users make fast and insightful pruning decisions that improve model performance on test data when there is a distribution shift. Notably, these decisions can be made even when the user knows little about the model architecture and has no knowledge of the true

underlying spurious correlation in the training data. We conduct a user study where we explore the benefits of human-guided pruning on PCBM and PCBM-h performance.

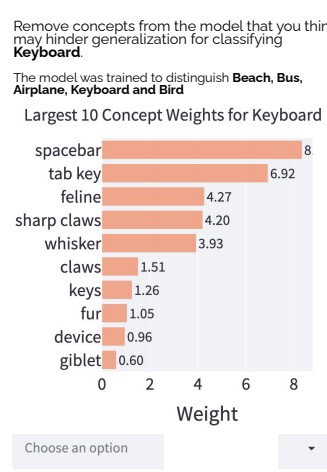

Figure 3: **User Study Interface.** We train PCBMs on MetaShift scenarios, each with a distribution shift between the training and test datasets. The user selects concepts to prune from the model.

**Study Design:**
We construct 9 MetaShift scenarios with a distribution shift for one of the classes between the training and testing datasets, which we refer to as the "shifted class". For instance, all of the training images for the class *Keyboard* also contain a *Cat* in the image, whereas test images do not have a cat in the image. The list of the 9 scenarios can be found in the Appendix. For each scenario, we train a PCBM and a PCBM-h with CLIP concepts on the training dataset and task the users with making edits to the trained models by selecting concepts to prune. We use CLIP's ResNet50 variant as our backbone, and leverage ConceptNet to construct the concept subspace.

Figure 3 shows the concept-selection interface. We refer to the Appendix for the full set of instructions given to the user. The display consists of the classification task the model was trained on and the concepts with the ten most positive weights for the shifted class. The user selects a subset of the concepts to prune. Neither the original model accuracy nor the accuracy after pruning is revealed to the user until all scenarios are completed. The target audience for the study was machine learning practitioners and researchers; 30 volunteers participated for a total of $30 \times 9 = 270$ experiments. The IRB has determined that this does not require review.

**Human-guided editing is fast and improves model accuracy:**
On average, users prune $3.65 \pm 0.39$ concepts in $34.3 \pm 6.4$s per scenario. We evaluate the effectiveness of user pruning by comparing the model accuracy after user pruning to the accuracy of the unedited model on a held-out dataset from the test domain. All $30/30$ users improve the model accuracy when averaged over scenarios. $8/30$ users improve model accuracy in *all* scenarios and $26/30$ users improve model accuracy in at least 6 scenarios. We compare human-guided editing to three baselines:

1. **Random Pruning**: We uniformly select without replacement a subset of the top ten concepts to prune, matching the number of concepts pruned by the user for a fair comparison.
2. **Greedy Pruning (Oracle)**: We greedily select one of the top 10 concepts that, when pruned, improves the model accuracy the most. We match the number of concepts pruned by the user.
3. **Fine-tune (Oracle)**: The model is fine-tuned on samples from the test domain.

Table 4: **Human-guided editing improves model accuracy for PCBMs with CLIP concepts (N=30).** We show the test accuracy on the shifted class averaged across 9 MetaShift scenarios, before and after the model is edited. Accuracies and edit gains are reported as *mean ± standard error*. (For the pruning strategies, we first compute the mean test accuracy across users within each scenario, then calculate the overall mean and standard error from the within-scenario means). We see that user pruning surpasses random pruning and can achieve nearly 50% of the accuracy gains made by fine-tuning and over 80% of the accuracy gains made by greedy pruning.

|  | Unedited | Random Prune | User Prune | Greedy Prune (Oracle) | Fine-tune (Oracle) |
|---|---|---|---|---|---|
| PCBM Accuracy | $0.620 \pm 0.035$ | $0.604 \pm 0.039$ | $0.719 \pm 0.042$ | $0.740 \pm 0.041$ | $0.824 \pm 0.049$ |
| PCBM Edit Gain | — | -0.016 | 0.099 | 0.120 | 0.204 |
| PCBM-h Accuracy | $0.642 \pm 0.034$ | $0.622 \pm 0.037$ | $0.736 \pm 0.034$ | $0.766 \pm 0.034$ | $0.856 \pm 0.018$ |
| PCBM-h Edit Gain | — | -0.020 | 0.094 | 0.124 | 0.224 |

We consider greedy pruning and retraining to be oracles since they require "leaked" knowledge of the test domain. Table 4 reports the model performance improvements for PCBM and PCBM-h, averaged over all 270 experiments in the user study. Improvements within each separate scenario can be found in the Appendix. We note that even with the residual component in PCBM-h, we can still improve model performance by editing the concept bottleneck. We observe that user pruning achieves marked improvement in model performance compared to the unedited model, attaining around 50% of the accuracy gains from fine-tuning and 80% of the gains from greedy pruning.

## 5 RELATED WORKS

**Concepts** Using human concepts to interpret model behavior has been drawing increasing interest (Kim et al., 2018; Bau et al., 2017; 2020). Related work focuses on understanding if neural networks encode and use concepts (Lucieri et al., 2020; Kim et al., 2018; McGrath et al., 2021), or generate counterfactual explanations to understand model behavior (Ghandeharioun et al., 2021; Abid et al., 2022; Akula et al., 2020). Recent works evaluated the causal validity of explanations (Feder et al., 2021; Elazar et al., 2021; Goyal et al., 2019), e.g. to eliminate potential confounding effects. There is further increasing interest in automatically discovering the concepts that are used by a model (Yeh et al., 2020; Ghorbani et al., 2019; Lang et al., 2021). **Concept-based Models:** Concept bottleneck models (CBMs) (Koh et al., 2020) extend the earlier idea (Lampert et al., 2009; Kumar et al., 2009) of first predicting the concepts, then using concepts to predict the target. CBMs bring about interpretability benefits but require training the model using concept labels for the entire training dataset, which is a key limitation. CBMs have not been analyzed in terms of model edits. Recent work reveals that end-to-end learned CBMs encode information beyond the desired concept (Mahinpei et al., 2021; Margeloiu et al., 2021). Concept Whitening (Chen et al., 2020) aims to align each concept with an individual dimension in a layer. While a subset of dimensions is aligned, this approach lacks a bottleneck since there are dimensions that are potentially not aligned with a concept (comparable to PCBM-h). Barnett et al. (2021) learns prototypes and uses them as concepts, where the distance to prototypes determines the model prediction. PCBM-h is inspired by semiparametric models on fitting residuals (Härdle et al., 2004). PIE (Wang et al., 2021) has a similar approach, where they combine individual features with more complicated interaction terms for tabular data.

**Model Editing** Model editing aims to achieve the removal or modification of information in a given neural network. Several models investigated editing factual knowledge in language models: Zhu et al. (2020) use variants of fine-tuning to achieve this objective while retaining performance on unmodified factual knowledge and Mitchell et al. (2021); De Cao et al. (2021); Hase et al. (2021) update the model by training a separate network to modify model parameters to achieve the desired edit. Similarly, Sotoudeh & Thakur (2021) proposes "repairing" models by finding minimal parameter updates that satisfy a given specification. One thread of work focuses on intervening on the latent space of neural networks to alter the generated output towards the desired state, e.g. removal of artifacts or manipulation of object positions (Sinitsin et al., 2020; Bau et al., 2020; Santurkar et al., 2021). For instance, Bau et al. (2020) edits generative models. Santurkar et al. (2021) edits classifiers by modifying 'rules', such as making a model perceive the concept of a 'snowy road' as the concept of a 'road', and they achieve this by modifying minimal updates to specific feature extractors. FIND (Lertvittayakumjorn et al., 2020) prunes individual neurons chosen by users and later fine-tunes the model. Right for the Right Concepts (Stammer et al., 2021) similarly fine-tune a neuro-symbolic model with user-provided feedback maps. Bontempelli et al. (2021) give an overview of concept-based models and a discussion on debugging strategies while concurrent work ProtoPDebug (Bontempelli et al., 2022) proposes an efficient debugging strategy.

## 6 LIMITATIONS AND CONCLUSION

In this work, we presented Post-hoc CBMs as a way of converting any model into a CBM, retaining the original model performance without losing the interpretability benefits. We leveraged multimodal models as an interface to use concepts, without the laborious concept annotation steps. Further, in addition to the local intervention benefits of CBM, we demonstrated that PCBMs can be leveraged to perform global model interventions. Many benefits of CBMs depend on the quality of the concept library. The concept set should be expressive enough to solve the task of interest. Users should be careful about the concept dataset used to learn concepts, which can reflect various biases. While there are several such real-world tasks, it is an open question if human-constructed concept bottlenecks can solve larger-scale tasks(e.g. ImageNet level). Hence, finding concept subspaces for models in an unsupervised fashion is an active area of research that will help with the usability and expressivity of concept bottlenecks. Multimodal models provide an effective interface for concept-level reasoning, yet it is unclear what is the optimal way to have humans in the loop. Here, we presented a simple way of taking human input via concept pruning; how to incorporate richer feedback is an interesting direction for future work.

## ACKNOWLEDGMENTS

We would like to thank Duygu Yilmaz, Adarsh Jeewajee, Carlos Guestrin, Edward Chen, Kyle Swanson, Omer Faruk Akgun, and Roxana Daneshjou for their support and comments on the manuscript, and all members of the Zou Lab and Guestrin Lab for helpful discussions. We thank the anonymous reviewers for their suggestions to improve this manuscript. J.Z. is supported by NSF CAREER 1942926 and the Chan-Zuckerberg Biohub.

## ETHICAL STATEMENT

In our user study, we did not collect any user information. We applied to the IRB within our institution and "IRB has determined that this research does not involve human subjects as defined in 45 CFR 46.102(f) and therefore does not require review by the IRB".

## REPRODUCIBILITY STATEMENT

The code to reproduce all experiments will be released at `https://github.com/mertyg/post-hoc-cbm`. All of the backbones we used to construct concept bottlenecks are released public checkpoints, and we stated all of these in the manuscript.

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

## A  TRAINING DETAILS

**Hyperparameters:** In all of our experiments ElasticNet sparsity ratio parameter was $\alpha = 0.99$. We trained all our models on a single NVIDIA-Titan Xp gpu. All of the models were trained for a total number of 10 epochs. We tune the regularization strength on a subset of the training set, that is kept as a validation set. PCBMs are fitted using scikit-learn's SGDClassifier class, with 5000 maximum steps. Hybrid parts are trained with PyTorch, where we used Adam as the optimizer with 0.01 learning rate, with 0.01 $L_2$ regularization on the residual classifier weights, and trained for 10 epochs.

**Dataset Details:**

1. Metashift: For all metashift experiments, we have 5 class classification problems. We have 50 images for each class as the training set, and 50 images for each class in the test dataset, which is different from the training images. The regularization strength for Metashift experiments is 0.002.

2. CIFAR: We use the original training and test splits of CIFAR datasets. The regularization strength for CIFAR10 and CIFAR100 is $\frac{2.0}{KN_c}$. We use linear probing to evaluate the original model

3. COCO-Stuff: We use the original training and test splits of COCO dataset. We sample 500 training and 250 test images from the dataset for each class, and we upsample the images whenever there are not enough images. Recognizing each of the 20 biased classes itepsingh2020don is treated as a binary classification task where we minimize binary cross entropy loss separately for 20 classes, and compute the mean average precision metric. The regularization strength for COCO-Stuff is 0.001. We evaluate the original model performance using linear probes with CLIP.

4. Ham10k: We split 80% of the HAM10k dataset and use it as the training set, and use the remaining 20% as the test data. The regularization strength for Ham10k is $\frac{2.0}{KN_c}$.

5. ISIC: We use 2000 images for training (400 malignant, 1600 benign) and evaluate the model on a held-out set of 500 images (100 malignant, 400 benign). The regularization strength for ISIC is $\frac{0.001}{KN_c}$. We evaluate the original model performance using a linear probe.

6. CUB: We use the training and test splits provided in Koh et al. (Koh et al., 2020). The regularization strength for CIFAR10 and CUB is $\frac{0.01}{KN_c}$.

**Visual Concepts**: We used Broden Visual concept bank for CIFAR and controlled editing experiments, CUB's training data for CUB concepts, and derm7pt dataset for dermatology concepts. For each of these, we use 50 pairs of positive and negative images, and learn a linear SVM. We use the vector normal to the decision boundary as the concept vector.

**Multimodal concepts:** For concept learning, we leveraged the ConceptNet hierarchy. For each classification task, we searched concept net for the class name and obtained concepts that have the following relation with the query concept: *hasA*, *isA*, *partOf*, *HasProperty*, *MadeOf*. We share the code for obtaining natural language concepts using ConceptNet. For each of these concepts, we obtain the text embedding using CLIP and use those as the concept vector.

## B    RESIDUAL COMPONENT INTERVENES ONLY WHEN NECESSARY

Does PCBM-h alter predictions made by the PCBM? We hypothesized that the residual component would intervene only when the concept bottleneck is not sufficient. To better understand the effect of residual predictor, we analyzed the prediction consistency between the two models for CIFAR10 and CIFAR100, where we looked at the models with labeled concepts (i.e. 170 concepts). In Figure 4 and 5, x-axis denotes the confidence of PCBM, the orange line gives the accuracy for samples with the given confidence, and the blue line gives the consistency between PCBM and PCBM-h for the same samples, i.e. whether they make the same prediction. In CIFAR10 and CIFAR100, we show that PCBM and PCBM-h consistency is high when the model confidence is high, and the model accuracy is high. Consequently, PCBM-h changes the model prediction mostly when the PCBM prediction is likely a mistake, and the confidence is low. In Figure 5, we show that all of the predictions changed by PCBM-h are to fix model mistakes. Further, in Figure 6 we show the PCBM confidence and the mean absolute deviation between the PCBM confidence and the PCBM-h confidence. We observe that PCBM-h has less effect on model confidence as the model gets more confident.

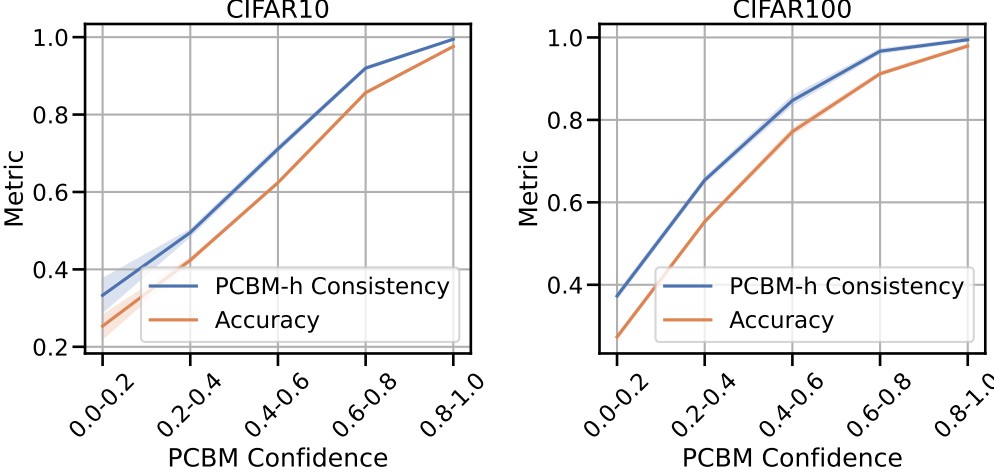

Figure 4: **Residual component intervenes mostly when the confidence is low.** Here, we look at the consistency between PCBM and PCBM-h predictions (i.e. whether both models make the same prediction). Namely, at each confidence level for the PCBM, we report the accuracy and consistency with the PCBM-h predictions. Overall, we see that PCBM-h is most likely to change the model prediction when the model is making a mistake, and otherwise, predictions are consistent.

## C    COMPARISON TO CBM

In most of our experiments, there is no clear way to run CBM as a benchmark, as none of the datasets have dense concept annotations - except CUB. In CUB, we run CBM as a baseline. Particularly, we try the joint training strategy from Koh et al. (Koh et al., 2020), where it was reported to

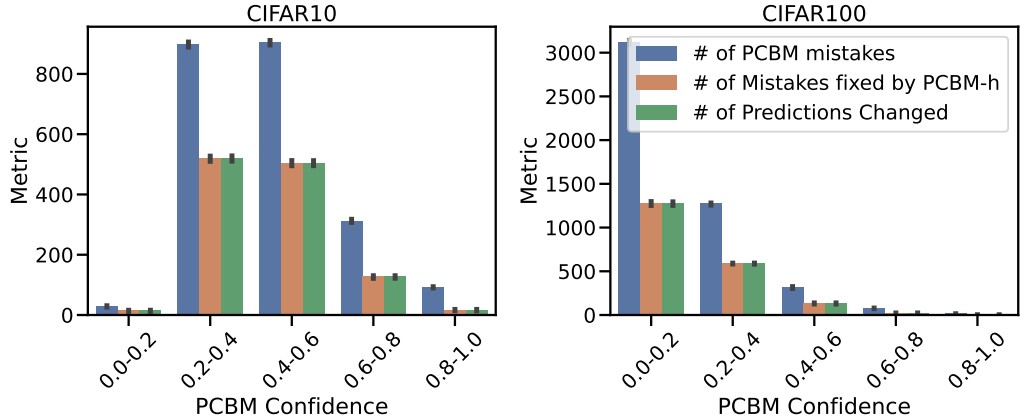

Figure 5: **Residual component intervenes only to fix mistakes.** Here we show the number of mistakes, the number of predictions changed by PCBM-h, and the number of mistakes fixed by PCBM-h. We see that PCBM-h only changes the model predictions to fix model mistakes.

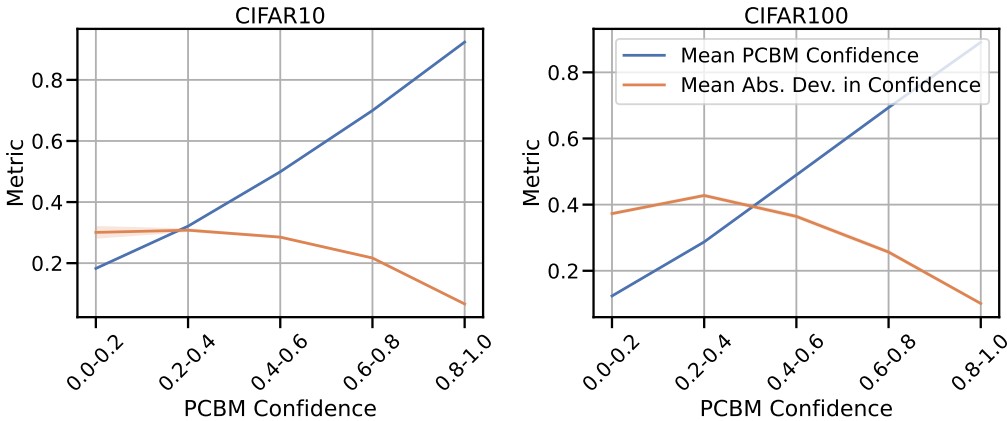

Figure 6: **Effect of the residual predictor on the confidence.** Here we show the PCBM confidence and mean absolute deviation between the PCBM confidence and the PCBM-h confidence. We observe that PCBM-h has less effect on model confidence as the model gets more confident.

give the best results in the CUB dataset. To make the comparison equal, we use the same set of concepts in both models, and use a linear predictor layer. Further, in both cases, we use the same frozen ResNet18 backbone. We searched over a grid of learning rates from $\{0.01, 0.01, 0.1, 1.0\}$ and $\lambda \in \{0.001, 0.01, 0.1, 1.0\}$ where $\lambda$ is the coefficient of the concept predictors in the joint training objective (see the CBM paper, Section 3), and the ElasticNet regularization strength in $\{0.001, 0.01, 0.1, 1.0\}$. In Table 5, we report the model performance. Overall, we observe some benefit from using dense concept annotations over the entire training dataset, where CBMs achieve a slightly better performance than PCBMs, and the original backbone. We note that this was at the cost of 112 concept annotations for each of the training samples.

We further analyze the behavior of CBM under different number of annotations. In Figure 7 we train the CBM with varying number of annotations. CBMs require dense annotations, e.g. 11200 annotations would mean $11200/112 = 100$ images. On the x-axis, we give the number of annotations used to train the CBM. On the y-axis, we give the accuracy on the test set. We observe that CBMs require a much larger amount of annotations to achieve the same accuracy as PCBMs.

Table 5: **Comparison to CBM.** We compare PCBM to CBM in the CUB dataset. The accuracy over the test set is reported.

|  | ResNet18(Backbone) | PCBM | PCBM-h | CBM(Backbone Fixed) |
|---|---|---|---|---|
| CUB Accuracy | 0.612 | 0.588 | 0.610 | 0.629 |

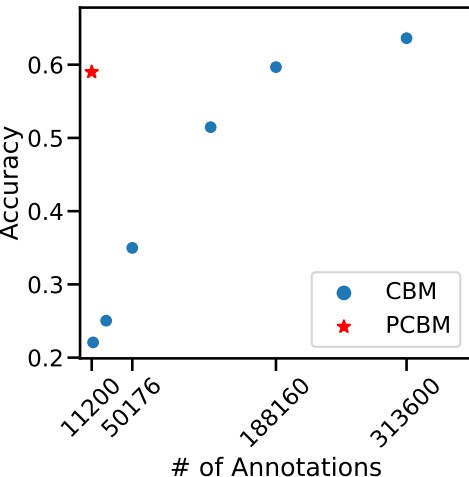

Figure 7: **Effect of the number of annotations in CBM.** Here we train the CBM with varying number of annotations. CBMs require dense annotations, e.g. 11200 annotations would mean $11200/112 = 100$ images. On the x-axis, we give the number of annotations used to train the CBM. On the y-axis, we give the accuracy on the test set. We observe that CBMs require a much larger amount of annotations to achieve the same accuracy as PCBMs.

## D    CONTROLLED METASHIFT EXPERIMENTS FOR MODEL EDITING

For Metashift, we have 2 tasks. Both tasks are 5-class object recognition tasks, where in the first one classes are *airplane, bed, car, cow, keyboard*, and for the second one we have *beach, computer, motorcycle, stove, table*. For each of these, we use a ResNet18 pretrained on ImageNet as the backbone of the P-CBM, and then use 100 images per class to train the concept bottleneck. For all experiments, we use the Adam Optimizer with a learning rate of $0.05$, the regularization parameters $\lambda = 0.05$, $\alpha = 0.99$. Similar to CIFAR experiments, we use the Broden Concept dataset. Below we give the entire set of results.

| Train | Test | Model | Original | Prune | Prune+Normalize | Fine-Tune |
|---|---|---|---|---|---|---|
| bed(dog) | bed(cat) | P-CBM | 0.760 | 0.760 | 0.760 | 0.920 |
| bed(cat) | bed(dog) | P-CBM | 0.680 | 0.700 | 0.720 | 0.940 |
| table(dog) | table(cat) | P-CBM | 0.520 | 0.540 | 0.620 | 0.760 |
| table(cat) | table(dog) | P-CBM | 0.660 | 0.700 | 0.740 | 0.760 |
| table(books) | table(dog) | P-CBM | 0.600 | 0.580 | 0.780 | 0.720 |
| table(books) | table(cat) | P-CBM | 0.620 | 0.680 | 0.800 | 0.820 |
| car(dog) | car(cat) | P-CBM | 0.718 | 0.718 | 0.744 | 0.949 |
| car(cat) | car(dog) | P-CBM | 0.620 | 0.760 | 0.840 | 0.840 |
| cow(dog) | cow(cat) | P-CBM | 0.778 | 0.750 | 0.778 | 0.944 |
| keyboard(dog) | keyboard(cat) | P-CBM | 0.620 | 0.580 | 0.720 | 0.940 |
| bed(dog) | bed(cat) | HP-CBM | 0.760 | 0.760 | 0.780 | 0.900 |
| bed(cat) | bed(dog) | HP-CBM | 0.760 | 0.740 | 0.760 | 0.940 |
| table(dog) | table(cat) | HP-CBM | 0.600 | 0.620 | 0.640 | 0.780 |
| table(cat) | table(dog) | HP-CBM | 0.540 | 0.580 | 0.640 | 0.820 |
| table(books) | table(dog) | HP-CBM | 0.660 | 0.700 | 0.760 | 0.680 |
| table(books) | table(cat) | HP-CBM | 0.760 | 0.800 | 0.820 | 0.780 |
| car(dog) | car(cat) | HP-CBM | 0.795 | 0.769 | 0.795 | 0.974 |
| car(cat) | car(dog) | HP-CBM | 0.640 | 0.660 | 0.740 | 0.720 |
| cow(dog) | cow(cat) | HP-CBM | 0.639 | 0.639 | 0.639 | 0.917 |
| keyboard(dog) | keyboard(cat) | HP-CBM | 0.400 | 0.460 | 0.560 | 0.940 |

Table 6: Results of the Metashift editing experiments.

# E    USER STUDY

## E.1    USER INTERFACE

Below, we provide screenshots of the initial launch page and the concluding summary page from the user study. For the concept-selection interface, refer to the main paper.

**Background**

Typical end-to-end deep learning models for image classification are difficult-to-interpret black boxes that map input pixels directly to prediction. A **Post-hoc Concept Bottleneck Model (PCBM)** enhances interpretability by making output predictions using human-understandable concepts. It first maps the image input to a "bottleneck" of concepts, that maps the concepts to predictions. A PCBM offers insight into what concepts the model "sees" in the input and what concepts it deems as "important" and "relevant" for making the prediction.

**Instructions**

We are giving you the power to *edit* a trained PCBM! You will be asked to perform edits for 9 scenarios. In each scenario, the model has been trained for a 5-class classification task. 1 of the 5 classes will be designated as a class of interest. You will be shown the 10 concepts our model relies on the most to predict this class of interest, i.e. the 10 concepts with the most positive weights.

Your job is to remove concepts from the model that might hinder generalization. In other words, **the goal of your edits is to improve the prediction accuracy** for the class of interest **in the real world** . For instance, it might make sense to zero out concepts that seem completely unrelated to the class. You may also use other rationales to choose which concepts to remove.

First Name

Last Name

Start

Figure 8: **Launch page for user study.** Before starting the editing tasks, participants are shown a launch page containing a brief background section on PCBMs and a set of instructions.

**Scenario 1 / 9**

*The model was trained to distinguish **Beach** , **Bus** , **Airplane** , **Keyboard** , and **Bird** .*

**Before removing concepts**                    **After removing concepts**

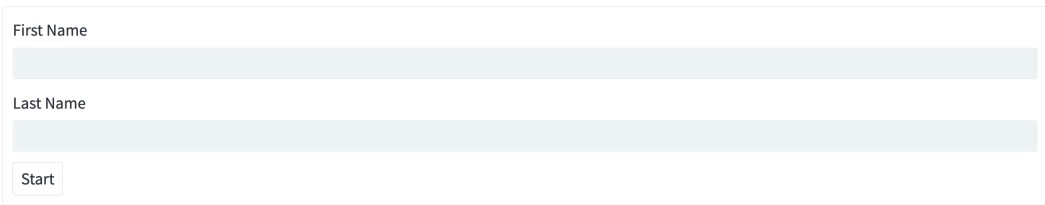

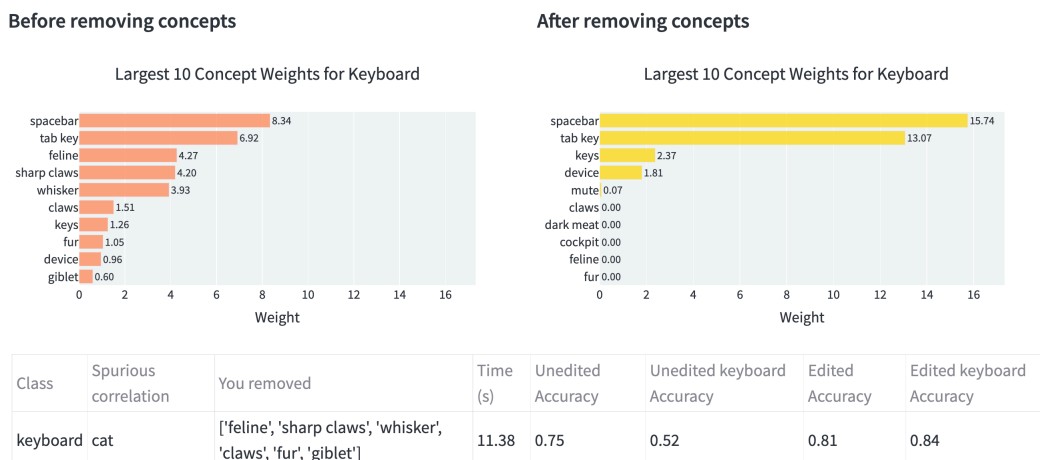

| Class | Spurious correlation | You removed | Time (s) | Unedited Accuracy | Unedited keyboard Accuracy | Edited Accuracy | Edited keyboard Accuracy |
|---|---|---|---|---|---|---|---|
| keyboard | cat | ['feline', 'sharp claws', 'whisker', 'claws', 'fur', 'giblet'] | 11.38 | 0.75 | 0.52 | 0.81 | 0.84 |

Figure 9: **Summary page for user study.** After completing all 9 scenarios, participants are shown a summary page that includes the user's choice of concepts, the accuracy of the unedited model, and the accuracy achieved by the edited model.

E.2    HUMAN-GUIDED PCBM EDITING: DETAILED PERFORMANCE RESULTS

In each of the 9 scenarios in the user study, the underlying classification task is a 5-class object recognition task, and one of the classes has a spurious correlation in the training set. For instance, all

training images for the class *keyboard* also contain a *cat* in the image. In Table 7, we list the classes in the underlying classification task and the spurious correlation for each scenario.

Table 7: **Classification tasks and spurious correlations in user study scenarios.**

| Classification Task | Spurious Correlation |
|---|---|
| airplane, bed, car, cow, keyboard | bed(dog) |
| beach, bus, airplane, keyboard, bird | keyboard(cat) |
| beach, car, airplane, bed, bird | bed(cat) |
| beach, motorcycle, airplane, couch, bird | couch(cat) |
| bus, painting, cat, computer, snowboard | painting(lamp) |
| bus, pillow, cat, computer, snowboard | pillow(clock) |
| bus, television, cat, computer, snowboard | television(fireplace) |
| car, fork, table, bed, computer | fork(tomato) |
| dog, car, airplane, couch, bird | car(snow) |

Below, we report the test accuracies for each of the 9 scenarios in our user study, both for the shifted class and for the overall classification task.

Table 8: **Model accuracy of PCBM-h with CLIP concepts after editing (N=30)** We report the test accuracy in the shifted class and the overall test accuracy for each individual scenario in the user study. Accuracy for the pruning strategies is averaged over users and is shown as *mean ± standard error*.

| Scenario | Unedited | Random Prune | User Prune | Greedy Prune | Fine-tune |
|---|---|---|---|---|---|
| | | Shifted Class Test Accuracy | | | |
| bed(dog) | 0.720 | 0.651 ± 0.023 | 0.832 ± 0.008 | 0.838 ± 0.008 | 0.940 |
| keyboard(cat) | 0.520 | 0.425 ± 0.040 | 0.788 ± 0.021 | 0.803 ± 0.015 | 0.940 |
| bed(cat) | 0.700 | 0.663 ± 0.016 | 0.789 ± 0.006 | 0.813 ± 0.008 | 0.860 |
| couch(cat) | 0.700 | 0.725 ± 0.033 | 0.763 ± 0.011 | 0.883 ± 0.010 | 0.960 |
| painting(lamp) | 0.640 | 0.617 ± 0.006 | 0.640 ± 0.006 | 0.659 ± 0.002 | 0.840 |
| pillow(clock) | 0.800 | 0.776 ± 0.008 | 0.815 ± 0.003 | 0.815 ± 0.003 | 0.820 |
| television(fireplace) | 0.480 | 0.485 ± 0.004 | 0.535 ± 0.008 | 0.564 ± 0.009 | 0.760 |
| fork(tomato) | 0.580 | 0.583 ± 0.012 | 0.655 ± 0.010 | 0.707 ± 0.009 | 0.840 |
| car(snow) | 0.640 | 0.676 ± 0.014 | 0.807 ± 0.013 | 0.817 ± 0.011 | 0.840 |
| | | Overall Test Accuracy | | | |
| bed(dog) | 0.864 | 0.850 ± 0.005 | 0.882 ± 0.001 | 0.882 ± 0.001 | 0.900 |
| keyboard(cat) | 0.764 | 0.741 ± 0.008 | 0.811 ± 0.004 | 0.814 ± 0.003 | 0.860 |
| bed(cat) | 0.820 | 0.808 ± 0.004 | 0.834 ± 0.001 | 0.841 ± 0.002 | 0.832 |
| couch(cat) | 0.840 | 0.839 ± 0.005 | 0.851 ± 0.002 | 0.858 ± 0.002 | 0.904 |
| painting(lamp) | 0.852 | 0.841 ± 0.001 | 0.845 ± 0.001 | 0.854 ± 0.001 | 0.884 |
| pillow(clock) | 0.888 | 0.882 ± 0.002 | 0.891 ± 0.001 | 0.891 ± 0.001 | 0.888 |
| television(fireplace) | 0.816 | 0.817 ± 0.001 | 0.827 ± 0.002 | 0.832 ± 0.002 | 0.856 |
| fork(tomato) | 0.700 | 0.695 ± 0.002 | 0.706 ± 0.001 | 0.711 ± 0.001 | 0.732 |
| car(snow) | 0.784 | 0.791 ± 0.003 | 0.816 ± 0.002 | 0.817 ± 0.002 | 0.848 |

## F ANALYSIS ON THE COCO-STUFF BIASES

Singh et al (Singh et al., 2020) identifies co-occurence biases in the COCO-Stuff dataset, where 20 categories frequently co-occur with other identified categories. In Table 10, reader can find the concepts, taken from Singh et al. In Table 10, we report the Top-5 concepts for the PCBM trained with CLIP concepts. In the Biased Context column, we give the category that co-occurs frequently with the given context. In PCBM Top-5 Concepts column, we give the concepts that have the highest

Table 9: **Model accuracy of PCBM with CLIP concepts after editing (N=30)** We report the test accuracy in the shifted class and the overall test accuracy for each individual scenario in the user study. Accuracy for the pruning strategies is averaged over users and is shown as *mean ± standard error*.

| Scenario | Unedited | Random Prune | User Prune | Greedy Prune | Fine-tune |
|---|---|---|---|---|---|
| | | | Shifted Class Test Accuracy | | |
| bed(dog) | 0.720 | 0.647 ± 0.024 | 0.830 ± 0.008 | 0.832 ± 0.008 | 0.900 |
| keyboard(cat) | 0.520 | 0.412 ± 0.040 | 0.787 ± 0.021 | 0.799 ± 0.016 | 0.980 |
| bed(cat) | 0.700 | 0.677 ± 0.017 | 0.813 ± 0.008 | 0.804 ± 0.007 | 0.820 |
| couch(cat) | 0.680 | 0.701 ± 0.034 | 0.751 ± 0.012 | 0.882 ± 0.011 | 0.920 |
| painting(lamp) | 0.580 | 0.579 ± 0.004 | 0.607 ± 0.007 | 0.635 ± 0.005 | 0.520 |
| pillow(clock) | 0.740 | 0.760 ± 0.008 | 0.811 ± 0.005 | 0.799 ± 0.006 | 0.720 |
| television(fireplace) | 0.420 | 0.428 ± 0.004 | 0.459 ± 0.006 | 0.483 ± 0.007 | 0.880 |
| fork(tomato) | 0.600 | 0.581 ± 0.012 | 0.642 ± 0.007 | 0.685 ± 0.006 | 0.960 |
| car(snow) | 0.620 | 0.651 ± 0.014 | 0.777 ± 0.011 | 0.737 ± 0.008 | 0.720 |
| | | | Overall Test Accuracy | | |
| bed(dog) | 0.860 | 0.844 ± 0.005 | 0.878 ± 0.001 | 0.877 ± 0.001 | 0.880 |
| keyboard(cat) | 0.748 | 0.727 ± 0.008 | 0.802 ± 0.004 | 0.804 ± 0.003 | 0.800 |
| bed(cat) | 0.804 | 0.801 ± 0.004 | 0.830 ± 0.002 | 0.829 ± 0.001 | 0.668 |
| couch(cat) | 0.828 | 0.826 ± 0.006 | 0.841 ± 0.002 | 0.850 ± 0.001 | 0.892 |
| painting(lamp) | 0.836 | 0.835 ± 0.001 | 0.841 ± 0.001 | 0.847 ± 0.001 | 0.840 |
| pillow(clock) | 0.876 | 0.879 ± 0.002 | 0.890 ± 0.001 | 0.888 ± 0.001 | 0.848 |
| television(fireplace) | 0.800 | 0.802 ± 0.001 | 0.808 ± 0.001 | 0.813 ± 0.001 | 0.812 |
| fork(tomato) | 0.704 | 0.699 ± 0.002 | 0.710 ± 0.001 | 0.705 ± 0.002 | 0.720 |
| car(snow) | 0.768 | 0.776 ± 0.003 | 0.799 ± 0.002 | 0.792 ± 0.002 | 0.732 |

| Biased Category | Biased Context | PCBM Top-5 Concepts |
|---|---|---|
| cup | dining table | table, coaster brake, crockery, dining table, column |
| handbag | person | bag, platform, woman, person, toiletry |
| apple | fruit | apple tree, fruit, edible fruit, citrus fruit, produce |
| car | road | traffic circle, intersection, car window, car mirror, trunk |
| bus | road | public transport, transportation, tube, carriages, traffic circle |
| potted plant | vase | tall plant, patio, vase, plant, plant organ |
| spoon | bowl | eating utensil, utensil, cutlery, crockery, flatware |
| microwave | oven | kitchen, washing machine, kitchen appliance, oven, countertop |
| keyboard | mouse | computer, computer system, portable computer, computer brand, mouse |
| clock | building-other | clock face, timepiece, the dial, plank, chimney |
| hair drier | towel | shower, bathroom, toiletry, plumbing fixture, toilet |
| skateboard | person | paved surface, person, outsole, instep, footwear |

Table 10: COCO-Stuff Concepts

weight for the given category. Looking at PCBM categories, we can see that we can identify the biased contexts. For instance, *cup* category has *table, dining table*, *apple* has *fruit, fresh vegetables*, *car* has *traffic circle, intersection* in the concepts identified as important, which are parallel to the biased context. In other cases, our model surfaced different potential co-occurence biases, such as *chimney* for *clock*, and *toilet* for *hair drier*. One important limitation for this analysis is that the desired concepts should exist in the concept bank. To further improve the pipeline, automatic concept discovery approaches shall be a fruitful research direction (Ghorbani et al., 2019; Yeh et al., 2020).

