# OpenReview forum: "Post-hoc Concept Bottleneck Models"
_ICLR.cc/2023/Conference — ICLR 2023 notable top 25%_

### Official Review · Reviewer_cGSM · 2022-10-15

**Confidence:** 3
**Correctness:** 3
**Technical Novelty And Significance:** 3
**Empirical Novelty And Significance:** 3
**Recommendation:** 8

**Clarity, Quality, Novelty And Reproducibility:**

This is a high-quality paper that is clearly written. It proposes a novel approach that extends upon several well-known and widely used methods. The experimentation seems thorough and conclusive, and the methods presented seem flexible and useful in a wide array of problems.

Some minor issues in the paper:
Typo:

PCBMs more accessible **in** various settings.

Missing references:

1. Explaining Classifiers with Causal Concept Effect (CaCE)
2. CausaLM: Causal model explanation through counterfactual language models
3. Amnesic probing: Behavioral explanation with amnesic counterfactuals


**Strength And Weaknesses:**

Strengths:

It is an extremely well-written paper on an important and timely problem, generating post-hoc concept-based explanations for trained neural networks. It clearly describes the limitations of existing methods and builds upon them in an elegant way, leading to a superior solution. The experiments are thorough and seem convincing, clearly demonstrating the usefulness and flexibility of the proposed method.

Weaknesses:

While I am generally supportive of the paper and its claims, I do find the lack of discussion around the causal validity of the generated concepts missing. There have been many studies in the literature highlighting the causal nature of post-hoc model explanations, which the authors do not address. Please see missing references at the next section (Clarity, Quality, Novelty And Reproducibility).



I understand that this is not the goal of this paper, but addressing the fact that the generated concepts and their estimated effect might not faithfully reflect their true causal effect on the trained model sounds important to me and should be a part of this paper.


**Summary Of The Paper:**

This paper proposes a method for creating a concept bottleneck model from any given model. The authors claim that the proposed approach does not sacrifice the performance of the model, and retains the interpretability benefits of concept bottleneck models along with easy model editing.

**Summary Of The Review:**

It is an interesting paper on an important and timely problem, generating post-hoc concept-based explanations for trained neural networks. It proposes a novel approach that extends upon several well-known and widely used methods. The experimentation seems thorough and conclusive, and the methods presented seem flexible and useful in a wide array of problems. However, the lack of discussion around the causal validity of the generated concepts is missing.

---

> ### Author Response · Authors · 2022-11-08
> **Author Response to Reviewer cGSM**
>
> Dear Reviewer cGSM,
>
> Thank you for your detailed review and kind comments! We appreciate your kind words on the paper. It is evident that you spent a substantial time reviewing our paper, and we are thankful for your efforts during the 2 week review period. We are excited that you found the paper high quality, extremely well-written, our problem timely and the solution to be novel.
>
> We appreciate your comments on causal validity, and we agree that we should mention faithfulness to the underlying causal model and interactions between concepts. We added references to this to the manuscript, along with an update on the suggested typo and missing citations (See the end of section 3, and see the updated related works, marked with blue).
>
> Overall, we appreciate your comments and we believe that this helped make the story more complete. Thank you again for your kind review. Please let us know if you have further questions, we are happy to follow up if there are unaddressed points.

---

### Official Review · Reviewer_uQCo · 2022-10-24

**Confidence:** 4
**Correctness:** 3
**Technical Novelty And Significance:** 3
**Empirical Novelty And Significance:** 3
**Recommendation:** 8

**Clarity, Quality, Novelty And Reproducibility:**

Clarity: The paper is very well written.  There are some grammatical issues here and there, see for instance the abstract.  These can be easily fixed.

Quality: The proposed approach is quite sound.  I have some doubts that CAV vectors can fully capture the nuances of non-linear mappings, but given that they are widely accepted in the community, I do not have major issues with this modelling choice.  PCBM-h's are also sensible.  The authors clearly mention that these are affected by limited interpretability, and mention possible solutions (e.g., adapting ideas from the PIE paper).

The experiments themselves are quite extensive, however they neglect compakeyring PCBMs against one obvious baseline, namely Concept Whitening (which is however referenced in the paper), which is perhaps the technique that is closest in spirit to CBMs (they are actually quite similar, modulo an extra orthogonality constraint).  Adding this comparisong would strengthen the message substantially.  In my opinion, however, this is not a strict requirement.

My main issue with the text is the "Post-hoc concept bottlenecks do not hurt the full model performance" paragraph in p. 5.  Looking at Table 1 very clearly shows that performance *is* hurt, and only PCBM-h's manage to get close to the accuracy of the black-box model.  Here are the numbers (black-box -> PCBM) for the different data sets:

 CIFAR10: 0.888 -> 0.777 (*-11%*)

 CIFAR100: 0.701 -> 0.520 (*-18%*)

 COCO-Stuff: 0.770 -> 0.741 (3%)

 CUB: 0.612 -> 0.588 (2%)

 HAM10000: 0.963 -> 0.947 (2%)

 ISIC: 0.812 -> 0.736 (*8%*).

In three data sets out of six, there is a drop in performance > 5%.  It is debatable whether this is significant or not.  Now, I am perfectly fine with there being an (even noticeable) drop in performance, as long as this is reported correctly in the text.  To be clear, all I am explicitly asking  is that that the authors rephrase **the title of the paragraph** to better match what the results show.

Novelty: PCBMs combine ideas from CAVs and concept-based models in a novel way.  I expect this model to be of interest to CBM and XAI specialists.

Reproducibility: The code is available and seems well structured.  Pointers to datasets are provided.

**Strength And Weaknesses:**

PROS:

- PCBMs mix existing techniques into a novel recipe in a sensible manner.

- PCBMs overcome one of the main limitations of CBMs, namely the need for dense annotations.

- The manipulation experiment adds some flavour to the main message.

- The text is very well written and easy to follow.

CONS:

- Empirical results do not entirely match the claims in the text.

- Missing comparison with non-CBM baselines.

**Summary Of The Paper:**

The authors introduce a new class of interpretable neural network classifiers, denoted Post-hoc Concept Based Models (PCBMs).  PCBMs assume a concept vocabulary is given, fit linear representations of the various concepts in the embedding space of the target model using auxiliary positive and negative examples of each concept, then acquire an interpretable classifier on top of these embeddings (or rather the projection of the input on them).  The key advantage of PCBMs over regular CBMs is that the target data set needs not be densely annotated.  The authors also introduce a partially interpretable version of PCBMs (PCBM-h's) that upgrade performance by also making use of possibly uninterpretable features.  Both classes of models are compared on several data sets in terms of accuracy (against a black-box baseline) and amount of supervision required.

**Summary Of The Review:**

Simple but useful new class of models that overcome major issue with prior work, with a couple of key issues that can be easily fixed.

---

> ### Author Response · Authors · 2022-11-08
> **Author Response to Reviewer uQCo**
>
> Dear Reviewer uQCo,
>
> Thank you for your detailed review! It is evident that you spent a substantial time reviewing our paper, and we are thankful for your efforts during the 2 week review period. We appreciate your kind comments that our recipe is novel, our text is well written and our work could be of interest to CBM and XAI specialists.
>
> **Grammatical issues**: Thank you for pointing these out, and we apologize for the inconvenience. We tried to fix the grammar errors in the paper.
>
> **Claims and experimental results**: We understand this point. We considered PCBM-h as part of the post-hoc concept bottlenecks family, yet we see how it could give the impression that we are mentioning the PCBM model without the hybrid, and can appear as an overstatement. We tried to update the manuscript to fix this claim broadly, concretely with the change: “do not hurt model performance” -> “achieve comparable performance to the original model”. Please let us know if you have further suggestions on this front, and thank you for helping us better reflect the contributions of our work.
>
> **Comparison to Concept Whitening (CW)**: CW is a novel approach to building interpretable models. We cited CW as it is a post-hoc transformation of a black-box model to a model with interpretable dimensions. However, there are several important differences between CW and PCBM:
> - No bottleneck in CW: A key difference is that CW $\mathbb{R}^d \to \mathbb{R}^d$ is a mapping in the latent space, where the first $k$ dimensions are optimized to align with concepts. The rest of the $d-k$ dimensions are uninterpretable. It is unclear what the network is using beyond the first $k$ dimensions: There is no bottleneck (unless $d=k$, which is an impractical assumption). In this sense, CW is comparable to PCBM-h, but not to PCBM.
> - Concept interpretation:  PCBM's concept importance measure is natural: Coefficients of the linear model (exact effect of a concept on the class logits). CW proposes a heuristic measure by permuting the concept values for all samples and computing the sensitivity of the loss, lacking a direct interpretation.
> - Computational Complexity: CW optimizes the backbone, CW module, and the classifier, with multiple forward and backward passes for all concept images; multiple times in the epoch. PCBM uses 1 forward pass for each image and then learns the linear classifier.
> - Data Efficiency is an important consideration for label-scarce applications where concept annotations are expensive to obtain, e.g. medical use cases. PCBM needs only 50 (or fewer) pairs of images. CW uses the entire COCO dataset, and it is not clear whether it is possible to use smaller datasets.
> - Editing: Editing properties of CW have not been studied.
> Adding these differences together, we believe that our approach addresses different and complementary use cases compared to CW. We added more on CW to the updated paper.
>
> Overall, we appreciate your comments and we believe your comments helped us better reflect our contributions. Thank you again for your kind review. Please let us know if you have further questions, we are happy to follow up if there are unaddressed points.

---

> > ### Comment · Reviewer_uQCo · 2022-11-15
> > **Reply**
> >
> > Hi,
> >
> > apologies for the late reply.
> >
> > **Claims**: I'd prefer something along the lines of "PCBM perform well and PCBM-h's do not hurt performance", but I realize that it is tricky to convert this into an appealing paragraph description.  So I am okay with the proposed change.
> >
> > **Comparison to CW**: I am aware of these differences, and I am fine with not having a direct experimental comparison.  As I mentioned in my review, this is not a huge issue for me.

---

> > > ### Comment · Reviewer_uQCo · 2022-11-18
> > > **Further reply**
> > >
> > > Upon further reflection, I think this paper is above the "weak accept mark". The proposed model is contributing something useful that goes beyond what regular CBMs do, and most of its key limitations are clearly stated.  I have decided to increase the score accordingly.

---

### Official Review · Reviewer_n4tz · 2022-10-25

**Confidence:** 4
**Correctness:** 4
**Technical Novelty And Significance:** 4
**Empirical Novelty And Significance:** 4
**Recommendation:** 8

**Clarity, Quality, Novelty And Reproducibility:**

The work is clear and novel, and if the code is provided seems like it should be reproducible ( though i couldn’t check repo as it was redacted and not included as supplemental )

**Strength And Weaknesses:**

**Strength:**
The authors demonstrate the utility of their method for constructing CBMs in a post-hoc fashion using Concept net & CLIP if need be along with their model variant which uses a residual layer PCBM-h to get near same accuracy as the original model  ( at the expense of interpretability, but still a novel and simple addition to the architecture that is quite useful.

**Weaknesses:**
The paper had a few areas that could have been clearer (1) how to select negatives when given concept supervision and (2) it took me until after reading section 3 to understand when Visual or Multimodal concepts are used ( maybe that could be added to Figure 1?) You all mention it in your limitations section, but the use of Concept Net will not translate well to domain specific tasks or tasks with classes not in ConceptNet.

Finally on Page 6 before Figure 2, you mention PCBM-h and PCBM provide the same concept interpretations, but this is not exactly the same interpretation since the final prediction in PCBM-h is no longer a linear combination of concept weights, but rather concept weights and residual weights so  although the concept bottleneck learns the  same weights, the overall model behavior is different and the concepts influence in each model differ.



**Summary Of The Paper:**

The authors present Post-hoc Concept Bottleneck Models (PCBMs) as a method for converting any model into a CBM with its interpretability benefits while retaining the original model performance ( by introducing a residual layer that circumvents the bottleneck and helps with the concept set isn’t rich enough ).   When there isn’t concept supervision for a task, they use CLIP and ConceptNet as a method to learn concepts based on task classes.  They also show that PCBMs can be leveraged to perform global model interventions, specifically to reduce spurious correlations in a contrived dataset.  Through a model-editing user study, they show that editing PCBMs via concept-level feedback can provide significant performance gains without using any data from a target domain or model retraining for tasks in new data distributions.


**Summary Of The Review:**

The authors demonstrate the utility of their method for constructing CBMs in a post-hoc fashion using Concept net & CLIP if need be along with their model variant which uses a residual layer PCBM-h to get near same accuracy as the original model  ( at the expense of intepretability, but still a novel and simple addition to the architecture that is quite useful.

The paper mostly clearly written, though it took me a bit to understand certain parts such as when ConceptNet and Clip were used ( ie, when labeled concept examples don’t exist ).  The experiments well done and model editing was explained
well for the use case of datashift ( which is relatively simplified ), but still has utility.  The human user study was well done and very convincing.  It’d be interesting to explain the differences between the fine-tune oracle and user pruned concept weights in that section to see if fine tuning is actually picking up spurious correlations which are useful for task accuracy or as a way to suggest concepts for a human to consider?

---

> ### Author Response · Authors · 2022-11-08
> **Author Response to Reviewer n4tz**
>
> Dear Reviewer n4tz,
>
> Thank you for your detailed review! It is evident that you spent a substantial time reviewing our paper, and we are thankful for your efforts during the 2 week review period. We are also very excited that you found our approach quite useful, our experiments well done, and our human study convincing.
>
> **Clarifications**: We appreciate your help in identifying areas where we can explain better. In light of your suggestions, we added further clarification to the paper on how the concepts are learned and used.
> We updated our paper where we highlighted the updated text with blue. We highlighted the use of the concept datasets / multimodal models in Figure 1 and Table 1 and changed our statement on PCBM-h interpretations on Page 6 before Figure 2 to fix the claim.
>
> **On your comment regarding the oracle**: We think this is an interesting question for further investigation. One practical challenge is that a fine-tuning oracle is truly an oracle: Since we do not have access to the test distribution, we cannot finetune the model and inform the user on which concepts to consider. Thus in our experiments, this serves as a baseline for the best-case (albeit likely not possible to achieve) performance.
>
> We again thank you for your helpful feedback and time! Please let us know if there are any unaddressed points, we are happy to follow up.

---

### Decision · Program_Chairs · 2023-01-20

**Decision:**

Accept: notable-top-25%

**Justification For Why Not Higher Score:**

Need experiments with more pretrained models.

**Justification For Why Not Lower Score:**

The paper is worthy of being recognized as a spotlight based on its contributions.

**Metareview: Summary, Strengths And Weaknesses:**

Great, well written paper to convert a neural network to a concept bottleneck. The method is novel and the experiment results are convincing. All the reviewers agree that this paper should be accepted.

**Note From Pc:**

if the above contains the word "oral" or "spotlight" please see: "oral" presentation means -> notable-top-5% and "spotlight" means -> notable-top-25%. As stated in our emails, we are disassociating presentation type from AC recommendations